# Cross-Sectional Area and Echogenicity Reference Values for Sonography of Peripheral Nerves in the Lithuanian Population

**DOI:** 10.3390/diagnostics14131373

**Published:** 2024-06-28

**Authors:** Evelina Grusauskiene, Agne Smigelskyte, Erisela Qerama, Daiva Rastenyte

**Affiliations:** 1Department of Neurology, Medical Academy, Lithuanian University of Health Sciences, LT-44307 Kaunas, Lithuania; 2Department of Clinical Neurophysiology, Aarhus University Hospital, DK-8200 Aarhus, Denmark

**Keywords:** peripheral nerve ultrasound, Lithuanian references, normal values, nerve echogenicity, cross-sectional area

## Abstract

Objectives: We aimed to provide data of nerve sizes and echogenicity reference values of the Lithuanian population. Methods: High-resolution ultrasound was bilaterally performed according to the Ultrasound Pattern Sum Score and Neuropathy ultrasound protocols for healthy Lithuanian adults. Cross-sectional area (CSA) measurement and echogenicity were used as the main parameters for investigation. Echogenicity was evaluated using ImageJ, and nerves were categorized in classes according to echogenicity. Results: Of 125 subjects enrolled, 63 were males (mean age 47.57 years, range 25–78 years) and 62 were females (mean age 50.50 years, range 25–80 years). Reference values of nerve sizes and values of echogenicity as a fraction of black in percentage of cervical roots, upper and middle trunks of the brachial plexus and the following nerves: vagal, median, ulnar, radial, superficial radial, tibial, fibular, and sural in standard regions were established. Mild to moderate correlations were found between nerves CSA, echogenicity values and anthropometric measurements with the differences according to sex. Inter-rater (ICC 0.93; 95% CI 0.92–0.94) and intra-rater (ICC 0.94; 95% CI 0.93–0.95) reliability was excellent. Conclusions: Reference values of nerve size and echogenicity of Lithuanians were presented for the first time as a novel such kind of publication from the Baltic countries.

## 1. Introduction

High-resolution ultrasound (HRUS) is used to investigate the anatomical structure of peripheral nerves. Due to its relatively simple application and widespread availability, it became a field of interest more than a decade ago and has come a long way since. In 2021, the cross-sectional area (CSA) of some specific sites’ measurements were included in the European Academy of Neurology/Peripheral Nerve Society guidelines on diagnosis and treatment of chronic inflammatory demyelinating polyradiculoneuropathy as a supportive diagnostic criterion [1]. This signifies the current importance of HRUS as a diagnostic tool and its growing significance.

Another aspect of HRUS, peripheral nerve echogenicity, has also been analyzed [2,3]. However, most studies describe changes in nerve echogenicity in various polyneuropathy treatment trials [4] and leave a gap in the literature about echogenicity reference values in a healthy population.

Despite the fact that several studies have been performed in various world regions [5,6,7], it remains unclear whether or how HRUS measurements of peripheral nerves vary between ethnicities and age groups or according to different anthropometric indicators [7]. Therefore, it is essential to establish HRUS reference values for our own population. We hypothesize that a specific pattern of nerve size and echogenicity might exist in the Lithuanian cohort. Also, nerve size probably has an association with anthropometric parameters.

Reference measurements are needed in many areas of medicine, in order not only to make a more accurate diagnosis and improve treatment options, but also to collect data for scientific purposes and to analyze external factors that may have some influence. For the introduction of reference values, not only the specificities of different medical machines are important, but also the experience of each researcher.

Therefore, the aim of our study was to establish reference values of CSA and echogenicity of peripheral nerves in a Lithuanian cohort of healthy adults in association with demographic and anthropometric data.

## 2. Methods

This study was approved by the Kaunas Regional Biomedical Research Ethics Committee with bioethical permission No. BE-2–29, issued on 14 April 2022. A prospective study was conducted at the Department of Neurology of the Hospital of Lithuanian University of Health Sciences Kauno Klinikos (Kaunas, Lithuania) from April 2022 to October 2023.

### 2.1. Selection of Participants

In total, 125 healthy subjects with an age range of 25–80 years were recruited for this study. We followed recommendations given by the task force of the American Association of Neuromuscular and Electrodiagnostic Medicine and included more than 100 subjects in order to establish high-quality reference values [8]. Subjects from medical staff and patients who were treated in the neurology department for conditions that do not affect the peripheral nervous system (epilepsy, transient ischemic attack, stroke, and headache) were invited to participate. The purpose of the study was explained, and all participants gave written consent. Participants have not received any honorarium and participation was voluntary. All the participants were Caucasian and we matched the number of male and female subjects in the study: groups separated according to sex were categorized every ten years and >20 subjects in each category were recruited.

All subjects underwent a routine neurological examination. Data on age, sex, weight, height, and body mass index (BMI), were collected according to the World Health Organization recommendations [9]. Individuals with signs of polyneuropathy (weak reflexes, sensory disturbances), known neuropathies, diabetes or other endocrine disease, frequent alcohol consumption, history of neurotoxic drug use, malabsorption, and oncological diseases were excluded from the study. Considering the fact that subjects who were treated in the neurology department for other pathologies did not have any clinical signs of peripheral nerve system injuries, they were interpreted as healthy subjects, together with the subjects from the medical staff.

### 2.2. Ultrasound Examination

Ultrasound examination of peripheral nerves was performed using a Philips EPIQ 7 ultrasound machine (Philips EPIQ Diagnostic Ultrasound System, Bothell, WA, USA, 2019) with a linear 4–18 MHz transducer (eL18-4, piezo elements 1920) in B mode. The main presets of parameters of ultrasonic scanning were as follows: initial scanning depth 2.5 cm, dynamic range was 77. Presets were constant during the entire study and only depth was adapted when needed. Focus was kept on the area of interest all the time. The goal was to analyze two main parameters: nerve size and nerve echogenicity.

Ultrasound examination was performed by two specialists: a neurologist with 2 years of experience (EG) and a physician with 1 year of experience (AS) in neuromuscular ultrasound. First, all subjects underwent an ultrasound examination by the first investigator (EG). Then, twenty subjects [10] were remeasured in the same locations by the second investigator (AS) to establish reliable inter-examiner reproducibility. Lastly, investigator EG performed an ultrasound for the second time for the twenty subjects (a few weeks after the first measurement) to determine the intra-rater reproducibility. Both investigators performed CSA measurements in real-time. Investigators were blinded by each other’s results and EG was also blinded from the first measurement results. Images were anonymized and saved in the ultrasound machine for further echogenicity investigation.

We performed an ultrasound of the peripheral nerves and brachial plexus bilaterally and chose which side to start the examination in a randomized manner. The upper (UT) and middle trunks (MT) of the brachial plexus were measured in interscalene space, and the fifth (C5) and the sixth (C6) cervical roots were measured in two ways: perpendicularly (at the point where roots appear between the anterior and posterior tubercle of the transverse process) and longitudinally (just after leaving the transverse process). The vagal nerve (VN) was measured at the level of the carotid triangle. Upper limb nerves: median (MN) and ulnar (UN) measurements were recorded at the middle of the upper arm, elbow, and the middle of the forearm (the MN was also measured at the wrist and UN at Guyon’s canal), the radial nerve (RN) was measured at the spiral groove, and superficial RN was recorded at the arcade of Frohse. Lower limb nerves: tibial nerve (TN) and fibular nerve (FN) were measured at the popliteal fossa, and TN was measured at the ankle beneath the vascular arcade the sural nerve (SN) was measured next to the saphenous vein at the lateral ankle and in the calf. These measurement sites were chosen based on the Neuropathy ultrasound protocol (NUP) [11] and Ultrasound Pattern Sum Score (UPSS) [12] protocols. The size of the brachial plexus (except longitudinal view) and peripheral nerves were assessed by measuring the cross-sectional area. The CSA of the nerve was calculated automatically by tracing the circumference of the nerve along the hyperechoic epineurium (Figure 1). A tracer tool was used for all the CSA measurements, except cervical roots and brachial plexus trunks were measured using the ellipse function. For longitudinal images of cervical roots, a diameter measurement was used, measuring the distance between one side of the epineurium to the other.

To make echogenicity evaluation more objective, quantitative image analysis using ImageJ (National Institutes of Health, Bethesda, MD, USA), version 1.54d was performed. The same nerve boundaries were used for measuring CSA, avoiding white dots access into the area being measured. Then native images were converted into 8-bit images while ImageJ software transacted each pixel in a range between 0 (black) and 255 (white) (Figure 2). After that, the threshold function was used, and the hyperechoic volume or fraction of white was calculated automatically. The fraction of black was calculated by subtracting the fraction of white from 100 [2]. For echogenicity evaluation, only unilateral images were analyzed.

All evaluated peripheral nerves were divided into three classes by the fraction of black percentage:Fraction of black > 67%—a hypoechogenic nerve;Fraction of black 33–67%—a mixed hypo-/hyperechogenic nerve;Fraction of black < 33%—a hyperechogenic nerve.

### 2.3. Statistics

SPSS version 21 (SPSS, Inc., Chicago, IL, USA) and Microsoft Excel for Windows version 2311 (Microsoft, Redmond, WA, USA) were used to analyze data. All parameters were checked for normal distribution, using a single-sample Kolmogorov-Smirnov test. The independent sample t test was used for the comparison of height and some CSA sites between males and females. The Mann–Whitney U test was used for the comparison of age, weight, and the rest of CSA sites, as well as echogenicity estimates (fraction of black in percentage) between sexes. The Kruskal–Wallis test was used to analyze CSA differences in age groups. For each subject, the CSA for each nerve site was calculated as a mean of both sides (right + left/2). Data of all cohort CSA size values are presented as mean, median, and 95% confidence interval (CI). Spearman’s and Pearson’s tests were used to assess for linear correlations. The strength of correlation was defined as weak if r was less than 0.4, moderate if 0.4–0.6, and strong if >0.6 [13]. Intra-class correlation (ICC) was used to establish the inter- and intra-rater reliability. ICC values were interpreted as poor (ICC < 0.40), fair (ICC = 0.40–0.59), good (ICC = 0.60–0.74), and excellent (ICC = 0.75–1.0) [14]. The chi-squared test was used to detect differences in frequencies of echogenicity categories between sexes. Statistical significance was assumed when *p* < 0.05.

## 3. Results

### 3.1. Demographic Data for CSA Study

We enrolled 125 subjects (63 males and 62 females), of whom 101 were from medical staff and 24 subjects were treated at the neurology department (four subjects were treated for epilepsy, seven for transient ischemic attack, four because of stroke, and nine subjects investigated because of headache). The mean age of the group was 49.02 years (range 25–80 years). Mean values of height, weight, and BMI were higher in men. Characteristics of study subjects are summarized in Table 1.

### 3.2. Ultrasound Examination Results

A total of 5690 measurement sites were analyzed. Both inter-rater and intra-rater reliability of nerves CSA measurements were excellent (ICC 0.93; 95% CI 0.92–0.94, and ICC 0.94; 95% CI 0.93–0.95, respectively).

Male subjects had larger CSA compared to female at most measurement sites (Figure 3). Therefore, normative reference values are presented for males (Table 2) and females (Table 3) separately.

### 3.3. Nerve Size Correlation with Demographic and Anthropometric Parameters

Considering that nerve sizes differ according to sex, the correlation analysis between nerve CSA and anthropometric measurements was performed in males and females separately.

Correlations between C5 CSA and C5 diameter, as well as C6 CSA and C6 diameter were calculated. C5 CSA and C5 diameter positively moderately or mildly correlated both in men and women (r = 0.570, *p* < 0.001, and r = 0.398, *p* = 0.002, respectively). Also, C6 CSA and C6 diameter positively moderately or strongly correlated both in men and women (r = 0.516, *p* < 0.001, and r = 0.612, *p* < 0.001, respectively)).

Brachial plexus size was found to correlate with age in women since mild positive correlations between C5 CSA, C6 CSA, C5 diameter, and C6 diameter and age (r = 0.297, *p* = 0.026; r = 0.343, *p* = 0.01; r = 0.339, *p* = 0.007, and r = 0.281, *p* = 0.027) were found. On the other hand, no apparent differences were found in any of the measurement sites according to 10-year age groups.

The results showed that most of the upper limb nerves CSA in most measurement sites mildly or moderately correlated negatively with height in females and mildly or moderately positively with BMI in both sexes (Table 4). Also, it was found that CSA of all lower limb nerves in most measurement sites correlated mildly or moderately positively with BMI or weight predominantly in women. No apparent differences were found in most of the measurement sites according to BMI subclasses.

No correlations were found between the UT or MT of the brachial plexus and height, weight, or BMI neither in women nor men.

### 3.4. Nerve Echogenicity Results

Of 104 subjects enrolled in nerve echogenicity analysis, 50 were males (mean age 46.70 years, range 25–78 years) and 54 were females (mean age 49.19 years (range 25–80 years)), *p* = 0.468.

Fractions of black in percentage of VN, brachial plexus UT, MN at wrist and FA were lower in men compared to women (median 18.63% (min 0.31, max 74.16) and median 45.21% (min 0.57, max 91.82), *p* = 0.005; median 46.76% (min 0.22, max 95.80) and median 63.38% (min 0.48, max 97.53), *p* = 0.047; median 74.61% (min 17.60, max 98.99) and median 83.44 (min 49.04, max 99.07), *p* = 0.008 and median 34.89% (min 0.46, max 88.17) and median 47.14% (min 3.15, max 91.28), *p* = 0.032, respectively). The fraction of black in the percentage of SFN was lower in women compared to men (median 23.12% (min 0.69, max 81.61) and (median 54.04% min 4.50, max 93.82), respectively (*p* < 0.01). Since some echogenicity differences were found by sexes, normative echogenicity reference values were presented for males (Table 5) and females (Table 6) separately and further analysis for correlations with age and anthropometric measurements was performed according to sex.

Mild positive correlations were found between echogenicity of vagal nerve (r_s_ = 0.295, *p* = 0.037), FN at popliteal fossa (r_s_ = 0.331, *p* = 0.002), SN at lower leg (r_s_ = 0.362, *p* = 0.01), moderate correlations UT of brachial plexus (r_s_ = 0.445, *p* = 0.001), SFN nerve (r_s_ = 0.427, *p* = 0.002) and age in men. Moderate positive correlation of SN in calf (r_s_ = 0.407, *p* = 0.002), mild negative correlation of echogenicity of median nerve at wrist (r_s_ = −0.341, *p* = 0.012) and age were found in females only.

MN echogenicity in the upper arm mildly negatively correlated with weight (r_s_ = −0.277, *p* = 0.043) and BMI (r_s_ = −0.277, *p* = 0.043); also, MN echogenicity in elbow negatively correlated with weight (r_s_ = −0.306, *p* = 0.0024) in women. Height had the least correlations with nerve echogenicity and mild positive correlation was found between echogenicity of SFN (r_s_ = 0.319, *p* = 0.027) and height in men only.

The distribution of nerve echogenicity across echogenicity classes by sex is presented in Figure 4. Women were more likely to have a hypoechoic median nerve (class 1) at the forearm compared to men (*p* = 0.032). Values of fraction of black in percentage of UT of brachial plexus were more often corresponding to class 2 in women and more often corresponding to class 3 in men (*p* = 0.022). Also, it was found that according to the fraction of black percentage, echogenicity values in men had more mixed SFN nerves (class 2) than in women (class 3) (*p* = 0.02).

## 4. Discussion

In this study, we presented Lithuanian healthy subjects’ nerve sizes and echogenicity reference values of peripheral nerves. According to our knowledge, this is the first publication of its kind from the Baltic countries.

Several studies have been performed in Europe: twenty-three studies analyzing the upper limb, seven studies of the lower extremity, and five analyzing cervical nerve roots and vagal nerve, but none from the Baltic countries [5,6,7].

We provided normal reference values of sites which are suggested by well-known ultrasound protocols for polyneuropathies (Ultrasound Pattern Sum Score and Neuropathy ultrasound protocol) [11,12] and upper and middle trunks of brachial plexus additionally as a basis for further studies.

While published data are controversial [6,15], our study demonstrated a consistent size difference in most measurement sites between sexes. We suggest that separate values for men and women might be more accurate to use in daily clinical practice and especially in clinical trials.

We have performed nerve root measurements in two ways and the correlation between the fifth and sixth cervical root’s CSA mostly correlated with diameter values. This points to a choice of the most convenient measurement method according to the patient’s constitution.

Though we have noticed a number of similarities, some differences were found while comparing our data to the meta-analysis results. It has been shown that age influences nerve size in children [16]. A meta-analysis performed in a pediatric cohort showed that mean nerve CSA increases with age in the median, ulnar, radial, and tibial nerves. A different meta-analysis of nerve cross-sectional area reference values in an adult cohort did not find any age influence on nerve size, except a small possible effect of age on the median nerve at the wrist [5,6,7]. Our findings agree with the conclusions of this systematic review and meta-analysis as we have shown that age does not influence peripheral nerve size, except for the brachial plexus in the female subgroup. These findings allow us to speculate that nerve size changes while a child is growing and becomes stable when adulthood is reached.

The size of upper limb nerves showed an association with height in women and BMI in both sexes in the Lithuanian cohort. Data about nerve size and height association are controversial, and it seems that studies analyzing this association in different ethnic groups were mostly conducted in the Asian population [17,18,19].

Also, we have found that the size of lower limb nerves was associated with BMI and weight predominantly in women. CSA of the upper and middle trunks of the brachial plexus did not correlate with anthropometric data in any sex. We hypothesize that these findings might be associated with Lithuanian ethnicity’s anthropometric data specificity.

When comparing our results to a meta-analysis which analyzed 77 ultrasound studies on ulnar nerve CSA, we find that the measured ulnar nerve size is similar [20]. However, we have found differences of ulnar nerve CSA according to sex that were not described by this meta-analysis. Our results match meta-analysis data showing that ulnar nerve size may depend on BMI, as we found that most ulnar nerve measurements positively correlated with BMI.

According to our study, the median nerve at the carpal tunnel (pisiformis level), forearm, and upper arm levels correlates with weight and BMI selectively in the female group. We have found that the median nerve is smaller in women compared to men along its entire length. This statement agrees with the meta-analytical results which included 73 studies and stated that median nerve size values are higher in men than in women [21]. We could not state that age influences nerve size in the median nerve and this was a meaningful difference between the meta-analysis and our results. We have noticed that the median nerve is smaller along its length in the Lithuanian population compared to the meta-analysis with the given size numbers, which suggests that the approximate Lithuanian population wrist-to-forearm ratio is higher than the ratio of 1.5, which was suggested in a previous study [21].

A meta-analysis of CSA of the tibial nerve at the ankle level of 3295 subjects showed a positive correlation between age and tibial nerve size which was contrary to our results. In our study, we noticed that the tibial nerve at the ankle level correlates with weight and BMI, and there is a much stronger correlation with both weight and BMI to the tibial nerve at the popliteal fossa.

Another meta-analysis analyzing 2695 lower limb measurements [22] showed interesting results while comparing tibialis nerve size according to geographical regions. It was shown that healthy controls from Eastern Europe demonstrated larger tibial nerve CSA at the ankle level while Europeans had smaller tibial nerve size [23]. The Lithuanian cohort has smaller tibial nerves at the ankle compared to the Eastern European cohort which corresponds to the results of the European cohort. Unexpectedly, the tibial nerve CSA at popliteal fossa were larger in the Lithuanian cohort compared to both European geographical subgroups.

We have not measured the length of the arm or leg, but we hypothesize that some anthropometric features specific to Lithuanian ethnicity might exist. Ultrasound examination is not only inexpensive and well-tolerated but also relatively easy to learn [10,24]. The inter- and intra-reliability of the results were excellent in the present study, indicating that ultrasound is easy to perform and reliable to use, as was shown in other studies as well [25,26,27].

As invasive diagnostic methods, such as nerve biopsy, become rarely used, sono-histological studies are of growing importance. It was shown that different sonographic patterns exist in acute and chronic phases of peripheral neuropathies [28]. CSAs of the tibial and fibular nerves near the popliteal fossa from this study were comparable to those observed in an ex vivo study assessing the fibular and tibial divisions of the sciatic nerve with HRUS and histological cross-sections. These findings implicate the relevance of such studies in establishing baseline values and bridging the gap between basic and clinical knowledge [29]. Despite valuable studies conducted, there still is a lack of information about echogenicity changes in a healthy population. Gamber et al., 2020 [30] have shown that semiautomatic echogenicity measurement has excellent reliability. However, they did not establish cut-off values for healthy subjects because of the wide variability of the data. Similarly, even analyzing a sufficiently large cohort (104 subjects), we also demonstrated a wide range of echogenicity values which makes it difficult to establish accurate reference values.

Nerve echogenicity differences found between sexes let us assume that different patterns of echogenicity according to sex might exist. Also, additional differences between sexes appeared after nerve classification according to echogenicity values.

Echogenicity values have not shown many associations with anthropometric measurements, but it was found that nerve echogenicity becomes more hypoechoic with age.

We did not use any new methodologies or techniques in our study. On the contrary, we were trying to use the same protocols as previous studies to obtain comparable results. We believe that our study might have an important contribution to future studies analyzing if a disparity exists according to different ethnic groups and different world regions. Also, our study might have important implications for further nerve size studies analyzing gender differences and the impact of body anthropometric parameters on nerve size and echogenicity ultrasound measurement results. Further nerve echogenicity studies are needed for specific echogenicity pattern identification within a larger cohort.

Our study has several limitations. Firstly, a part of our cohort were patients treated in the neurology department, so they were not strictly healthy subjects. However, the pathologies they were hospitalized for did not damage the peripheral nerves. Also, these subjects underwent neurological examination, which was normal. Secondly, we calculated nerve size correlations with various anthropometric data except for correlations with the length of limbs. Thirdly, both investigators had a relatively short period of experience in neuromuscular HRUS. However, the inter-ratio and intra-ratio reliability was excellent. And lastly, we have not assessed the nerve fascicle’s size and the fascicle number which would have provided additional useful information next to CSA and echogenicity data.

Strengths: We have analyzed a large cohort of healthy Lithuanian subjects and made a substantial number of measurements which allowed us to establish high-quality nerve size reference values.

In conclusion, reference values of nerve size and echogenicity of the Lithuanian population are presented for the first time. Our study showed that HRUS measurements of nerve size obtained from a large Lithuanian cohort of healthy subjects are mostly similar to those published previously; however, some differences have been found which could show that features specific to the Baltic ethnicity might exist.

## Figures and Tables

**Figure 1 diagnostics-14-01373-f001:**
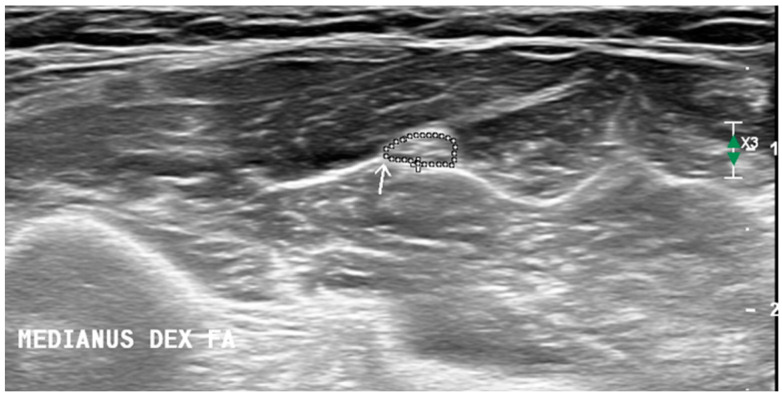
Cross-sectional area measurement methodology. Arrow is pointing to the median nerve at the forearm. CSA of the nerve is circled along the hyperechoic epineurium.

**Figure 2 diagnostics-14-01373-f002:**
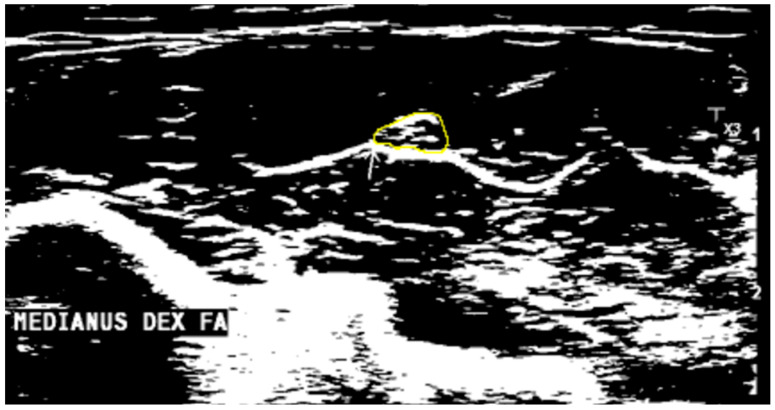
Arrow is pointing to the median nerve at the forearm. Image converted into an 8-bit image, each pixel in a range calculated between 0 (black) and 255 (white).

**Figure 3 diagnostics-14-01373-f003:**
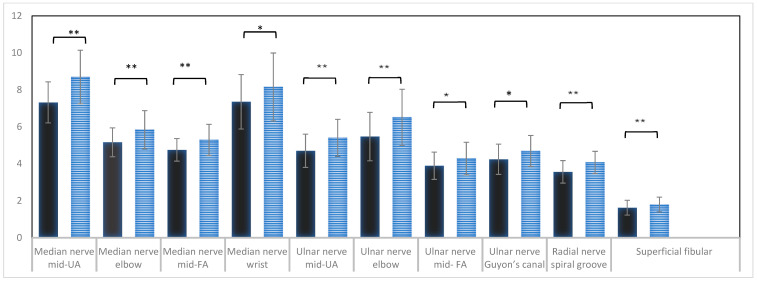
Overview over the significant findings among males and females. The figure shows the distribution of CSA of different nerves between males and females. The bars denote the mean and the SD for different measurement sites. Black bars denote females, and blue bars denote males. * *p* value < 0.05, ** *p* value < 0.001.

**Figure 4 diagnostics-14-01373-f004:**
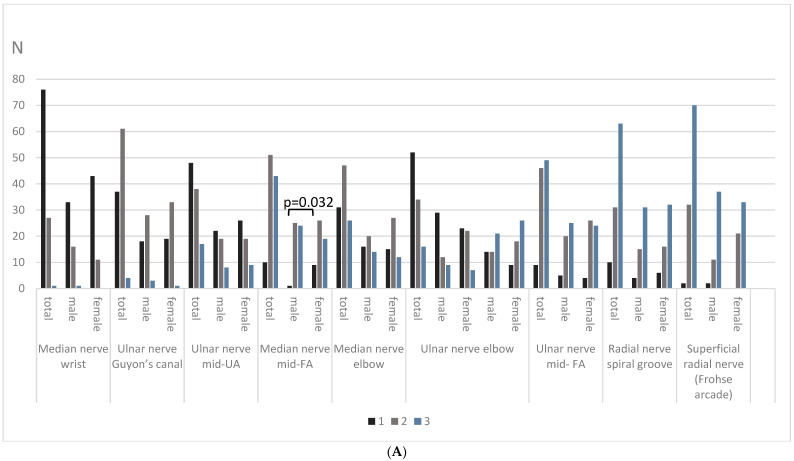
Distribution of nerve echogenicity classes according to gender ((**A**) upper limb, (**B**) lower limb, vagal nerve, and brachial plexus). (**A**) Mean, *p*-values for difference between echogenicity classes between males and females. Black bars denote echogenicity class 1, grey bars denote echogenicity class 2, blue bars denote echogenicity class 3, N—subject number. (**B**) Mean, *p*-values for the difference between echogenicity classes between male and female. Black bars denote echogenicity class 1, grey bars denote echogenicity class 2, blue bars denote echogenicity class 3, N—subject number.

**Table 1 diagnostics-14-01373-t001:** Demographic and clinical characteristics of the study participants for CSA analysis.

	All	Male	Female	*p*
Number	125	63	62	
Age range (years)	25–80	25–78	25–80	
Age (years) *	49.02 (16.83)	47.57 (16.96)	50.5 (16.72)	0.344
Height (cm) *	173.79 (9.10)	180.03 (7.19)	167.45 (5.93)	<0.01
Weight (kg) *	76.61 (14.56)	84.06 (12.70)	69.03 (12.29)	<0.01
BMI (kg/m^2^) *	25.31 (4.06)	25.93 (3.56)	24.68 (4.47)	0.036

* Mean (standard deviation).

**Table 2 diagnostics-14-01373-t002:** Reference values of peripheral nerve cross-sectional areas in males (*N* = 63).

Nerve Location	N (Sites)	Median (mm^2^)	Mean (mm^2^)	SD (mm^2^)	95% CI (mm^2^)
Cervical root 5	118	5.85	5.76	1.60	(5.34–6.17)
Cervical root 6	118	8.19	8.11	1.82	(7.64–8.59)
Cervical root 5 diameter	126	2.53 mm	2.57 mm	0.46 mm	(2.45–2.69) mm
Cervical root 6 diameter	126	3.42 mm	3.34 mm	0.57 mm	(3.20–3.48) mm
Vagal nerve	124	1.80	1.80	0.45	(1.69–1.91)
Median nerve mid-UA	126	8.62	8.70	1.45	(8.33–9.06)
Median nerve elbow	126	5.92	5.85	1.03	(5.59–6.10)
Median nerve mid-FA	126	5.06	5.29	0.84	(5.07–5.49)
Median nerve wrist	124	7.76	8.15	1.85	(7.68–8.62)
Ulnar nerve mid-UA	126	5.26	5.40	1.01	(5.15–5.66)
Ulnar nerve mid-FA	126	4.24	4.28	0.89	(4.06–4.50)
Ulnar nerve elbow	126	6.25	6.51	1.52	(6.13–6.89)
Ulnar nerve Guyon’s canal	126	4.72	4.70	0.84	(4.48–4.90)
Radial nerve spiral groove	126	4.11	4.07	0.61	(3.92–4.23)
Superficial radial nerve (Frohse arcade)	126	1.20	1.21	0.22	(1.15–1.26)
Tibial nerve popliteal fossa	124	24.96	25.90	5.19	(24.58–27.21)
Tibial nerve ankle	126	8.71	8.77	2.11	(8.23–9.29)
Fibular nerve popliteal fossa	124	4.84	4.82	0.86	(4.60–5.04)
Superficial fibular	124	1.69	1.79	0.41	(1.69–1.89)
Sural nerve distal calf next to SVV	126	1.52	1.54	0.34	(1.45–1.62)
Sural nerve calf	126	1.53	1.59	0.35	(1.51–1.68)
Upper trunk IT	124	4.88	5.04	1.37	(4.67–5.38)
Middle trunk IT	124	7.74	7.76	1.59	(7.36–8.16)

Mid—middle, UA—upper arm, FA—forearm, SSV—small saphenous vein, IT—interscalene space, SD—standard deviation, CI—confidence interval. Due to different subjects’ anatomical features, not all measurements could be performed in all sites, which determined the different site number (N) in distinct measurement sites. The mean of CSA for each nerve site was calculated as a mean of both sides (right + left/2). The given results show the cohort’s data as mean, median, and 95% confidence intervals.

**Table 3 diagnostics-14-01373-t003:** Reference values of peripheral nerve cross-sectional areas in females (*N* = 62).

Nerve Location	N (Sites)	Median (mm^2^)	Mean (mm^2^)	SD (mm^2^)	95% CI (mm^2^)
Cervical root 5	112	5.84	6.01	1.26	(5.67–6.35)
Cervical root 6	112	8.19	8.49	1.65	(8.05–8.93)
Cervical root 5 diameter (longitudinal)	124	2.57 mm	2.61 mm	0.38 mm	(2.52–2.71) mm
Cervical root 6 diameter (longitudinal)	124	3.38 mm	3.34 mm	0.51 mm	(3.22–3.47) mm
Vagal nerve	124	1.63	1.63	0.29	(1.56–1.70)
Median nerve mid-UA	124	7.25	7.32	1.11	(7.04–7.60)
Median nerve elbow	124	4.97	5.16	0.79	(4.96–5.36)
Median nerve mid-FA	124	4.79	4.75	0.61	(4.60–4.9)
Median nerve wrist	124	7.07	7.35	1.47	(6.97–7.72)
Ulnar nerve mid-UA	124	4.66	4.70	0.90	(4.47–4.93)
Ulnar nerve mid-FA	124	3.77	3.89	0.74	(3.70–4.08)
Ulnar nerve elbow	124	5.25	5.47	1.31	(5.14–5.80)
Ulnar nerve Guyon’s canal	124	4.24	4.25	0.83	(4.04–4.45)
Radial nerve spiral groove	124	3.59	3.56	0.61	(3.41–3.72)
Superficial radial nerve (Frohse arcade)	124	1.15	1.16	0.23	(1.10–1.22)
Tibial nerve popliteal fossa	124	22.49	22.74	5.32	(21.39–24.09)
Tibial nerve ankle	124	8.21	8.19	1.91	(7.7–8.68)
Fibular nerve popliteal fossa	124	4.70	4.80	1.08	(4.52–5.07)
Superficial fibular	122	1.55	1.62	0.40	(1.52–1.72)
Sural nerve distal calf next to SVV	124	1.53	1.63	0.43	(1.52–1.74)
Sural nerve calf	124	1.53	1.55	0.35	(1.46–1.64)
Upper trunk IT	120	4.92	4.90	1.12	(4.61–5.19)
Middle trunk IT	120	7.46	7.23	1.24	(6.91–7.55)

Mid—middle, UA—upper arm, FA—forearm, SSV—small saphenous vein, IT—interscalene space, SD—standard deviation, CI—confidence interval. Due to different subjects’ anatomical features, not all measurements could be performed in all sites, which determined the different site number (N) in distinct measurement sites. The mean of CSA for each nerve site was calculated as a mean of both sides (right + left/2). The given results show the cohort’s data as mean, median, and 95% confidence intervals.

**Table 4 diagnostics-14-01373-t004:** Nerve size correlation with anthropometric parameters.

	Males						Females					
	Height		Weight		BMI		Height		Weight		BMI	
Measurement Site	r/rs	*p*	rs	*p*	r/rs	*p*	r/rs	*p*	rs	*p*	r/rs	*p*
Cervical root 5	−0.012	0.93	0.147	0.267	0.158	0.232	-0.075	0.582	0.079	0.562	0.135	0.322
Cervical root 6	−0.007	0.956	0.036	0.784	0.078	0.557	−0.313 *	0.019	0.161	0.236	0.275 *	0.04
Cervical root 5 (longitudinal)	−0.132	0.304	0.075	0.56	0.177	0.166	−0.02	0.878	0.185	0.149	0.211	0.099
Cervical root 6 (longitudinal)	0.004	0.976	0.081	0.526	0.082	0.521	−0.214	0.095	0.275 *	0.031	0.373 *	0.003
Vagal nerve	−0.016	0.902	0.141	0.275	0.195	0.129	−0.276 *	0.03	0.095	0.463	0.269 *	0.035
Median nerve mid-UA	0.116	0.366	0.091	0.481	0.054	0.676	−0.267 *	0.036	0.181	0.158	0.286 *	0.024
Median nerve elbow	0.128	0.318	0.205	0.108	0.142	0.268	−0.316 *	0.012	0.073	0.574	0.215	0.093
Median nerve mid-FA	−0.104	0.104	0.326 *	0.009	0.280 *	0.026	−0.335 *	0.008	0.394 *	0.002	0.495 *	<0.001
Median nerve wrist	−0.051	0.692	0.173	0.179	0.188	0.143	−0.298 *	0.019	0.321 *	0.011	0.418 *	<0.001
Ulnar nerve mid-UA	0.138	0.282	0.326 *	0.009	0.292 *	0.02	−0.270 *	0.034	0.097	0.451	0.185	0.15
Ulnar nerve mid- FA	0.098	0.443	0.065	0.611	0.014	0.912	−0.415 *	<0.001	0.129	0.317	0.308 *	0.015
Ulnar nerve elbow	−0.069	0.589	0.267 *	0.035	0.251 *	0.047	−0.031	0.812	0.266 *	0.037	0.339 *	0.007
Ulnar nerve Guyon’s canal	0.113	0.378	0.128	0.316	0.045	0.726	−0.179	0.165	0.162	0.207	0.2	0.119
Radial nerve spiral groove	0.345 *	0.006	0.404 *	0.001	0.256 *	0.043	−0.116	0.37	0.18	0.162	0.252 *	0.048
Superficial radial nerve (Frohse arcade)	0.256 *	0.043	0.074	0.565	-0.057	0.655	−0.177	0.169	0.127	0.325	0.188	0.144
Tibial nerve popliteal fossa	0.134	0.298	0.537 *	<0.001	0.506 *	<0.001	0.01	0.939	0.492 *	<0.001	0.473 *	<0.001
Tibial nerve ankle	−0.025	0.843	0.094	0.466	0.086	0.501	−0.117	0.366	0.320 *	0.011	0.330 *	0.009
Peroneal nerve popliteal fossa	−0.056	0.663	0.16	0.215	0.23	0.072	−0.176	0.171	0.217	0.09	0.318 *	0.012
Superficial fibular nerve	0.106	0.41	0.13	0.314	0.059	0.649	−0.236	0.067	0.273 *	0.033	0.346 *	0.006
Sural nerve distal calf next to SVV	0.004	0.973	0.324 *	0.01	0.337 *	0.007	−0.18	0.161	0.15	0.245	0.233	0.068
Sural nerve calf	0.133	0.299	0.297 *	0.018	0.258 *	0.042	−0.135	0.296	0.105	0.418	0.164	0.203
Upper trunk IT	0.121	0.35	0.177	0.168	0.169	0.19	−0.042	0.748	0.042	0.748	0.085	0.519
Middle trunk IT	0.117	0.366	−0.005	0.967	−0.034	0.796	0	0.998	0.11	0.405	0.12	0.359

Mid—middle, UA—upper arm, FA—forearm, SSV—small saphenous vein, IT—interscalene space, BMI-body mass index. * Statistically significant.

**Table 5 diagnostics-14-01373-t005:** Reference values of echogenicity (fraction of black (%)) of peripheral nerves in males.

Site Name	N	Median (%)	Mean (%)	SD (%)	95% CI (%)
Vagal nerve	50	18.63	28.36	23.51	21.68–35.04
Upper trunk IT	50	46.76	43.10	36.25	32.80–53.40
Middle trunk IT	50	66.27	56.25	30.00	47.73–64.78
Median nerve wrist	50	74.61	73.50	19.22	68.04–78.97
Ulnar nerve Guyon’s canal	50	63.49	60.08	19.23	54.62–65.55
Median nerve mid-UA	49	64.52	58.77	26.15	51.26–66.28
Median nerve mid-FA	50	34.89	33.01	22.72	26.56–39.47
Median nerve elbow	50	60.29	50.54	28.43	42.46–58.62
Ulnar nerve elbow	50	69.10	60.75	27.24	53.01–68.49
Ulnar nerve mid-UA	49	46.35	39.91	29.03	31.57–48.25
Ulnar nerve mid- FA	50	32.87	33.63	24.12	26.78–40.49
Radial nerve spiral groove	50	12.60	27.51	26.37	20.02–35.00
Superficial radial nerve (Frohse arcade)	50	15.95	22.90	19.17	17.45–28.35
Tibial nerve popliteal fossa	49	49.69	36.57	25.44	29.27–43.88
Fibular nerve popliteal fossa	48	59.23	53.15	26.39	45.49–60.81
Superficial fibular	48	54.04	48.84	26.55	41.13–56.55
Tibial nerve ankle	50	55.87	52.72	20.01	47.03–58.40
Sural nerve calf	50	45.53	44.09	27.46	36.28–51.89
Sural nerve distal calf next to SVV	50	53.31	49.60	18.71	44.28–54.92

Mid—middle, UA—upper arm, FA—forearm, SSV—small saphenous vein, IT—interscalene space, SD—standard deviation, CI—confidence interval. Due to different subjects’ anatomical features, not all measurements could be performed in all sites, which determined different sites number (N) in distinct measurement sites.

**Table 6 diagnostics-14-01373-t006:** Reference values of echogenicity (fraction of black (%)) of peripheral nerves in females.

Site Name	N (Sites)	Median (%)	Mean (%)	SD (%)	95% CI (%)
Vagal nerve	54	45.21	42.60	25.47	35.65–49.55
Upper trunk IT	54	63.38	58.08	32.17	49.30–66.86
Middle trunk IT	54	73.42	61.93	28.51	54.15–69.71
Median nerve wrist	54	83.44	82.48	14.79	78.44–86.51
Ulnar nerve Guyon’s canal	54	62.45	63.18	12.62	59.74–66.62
Median nerve mid-UA	54	65.86	58.51	25.37	51.59–65.44
Median nerve mid-FA	54	47.14	43.26	22.94	37.00–49.53
Median nerve elbow	54	56.25	51.59	24.65	44.8–58.32
Ulnar nerve elbow	52	64.84	61.20	24.44	54.39–68.00
Ulnar nerve mid-UA	53	33.58	35.35	27.66	27.72–42.97
Ulnar nerve mid-FA	54	36.17	35.47	20.81	29.79–41.15
Radial nerve spiral groove	54	9.83	27.87	27.49	20.37–35.37
Superficial radial nerve (Frohse arcade)	54	21.67	27.04	21.16	21.26–32.82
Tibial nerve popliteal fossa	54	38.98	33.49	25.21	26.61–40.37
Fibular nerve popliteal fossa	53	58.16	54.13	26.31	46.88–61.38
Superficial fibular	50	23.12	29.42	25.50	22.17–36.67
Tibial nerve ankle	53	56.35	52.57	17.77	47.67–57.47
Sural nerve calf	53	35.42	36.10	25.19	29.16–43.05
Sural nerve distal calf next to SVV	54	51.79	49.15	20.37	43.58–54.71

Mid—middle, UA—upper arm, FA—forearm, SSV—small saphenous vein, IT—interscalene space, SD—standard deviation, CI—confidence interval. Due to different subjects’ anatomical features, not all measurements could be performed in all sites, which determined different sites number (N) in distinct measurement sites.

## Data Availability

The data presented in this study are available on request from the corresponding author.

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
