# Peer review of "Cross-Sectional Area and Echogenicity Reference Values for Sonography of Peripheral Nerves in the Lithuanian Population"

_diagnostics, 2024, doi:10.3390/diagnostics14131373_

Round 1

Reviewer 1 Report

Comments and Suggestions for Authors

Dear Authors, 

I was pleased to review the paper entitled " Cross-Sectional Area and Echogenicity Reference Values for Sonography of Peripheral Nerves in the Lithuanian Population" - 

- MDPI –

The present paper is very interesting, it focuses on a relevant clinical scenario, for orthopedics, potentially influencing the surgical and clinical practice for the management of peripheral nerves. 

Therefore, it is my opinion that the content is original, current, and relevant. 

Review editorial standards and format the text according to them.

Thus, there are some minor remarks:

- Title: The title gives a fine idea of the topic to be covered.

- Abstract: correct.

- Introduction: You can add pictures of the technique in this section.

Ultrasound had a significant increase in its use even during the period of the covid pandemic (you could add from MDPI doi: 10.3390/jcm11164785)

Conclude the introduction with your study hypothesis.

- Method: It better details the age range (line 64).

Specifies whether patients enrolled in the study received a fee or it was all for free.

Specify the inclusion and exclusion criteria more precisely. considering patients without a peripheral nerve disorder, indicate exactly the neurological disorders enrolled, if possible also express the data in a table.

Why did the second investigator evaluate only 20 subjects? I believe that all patients seen by the first should be reviewed by the second investigator as well!! The sentence is not clear, explain it.

Conclude the methods, before the statistical section, by stating exactly the endpoints of the study and therefore the hypotheses of the work.

- Results: correct.

- Discussion: The objectives of the work are not clear. What advantage to the scientific literature might your study add?

Conclude this section with limitations, strengths, future implications of your paper.

I recommend adding figures to increase the value of the research and making the methods part clear.

The paper generally is well written and needs only minor changes.

Comments on the Quality of English Language

Minor editing of English language required

Author Response

Dear reviewer,

Thank You so much for Your insights and notes. We found them very valuable. We are sending our corrections and addition information which was asked. 

  1. Introduction: You can add pictures of the technique in this section.

The image of technique measuring cross-sectional area and the example of image used for echogenicity measurement was included into sections "2. Methods" where methodology described in detail.

  1. Ultrasound had a significant increase in its use even during the period of the covid pandemic (you could add from MDPI doi: 10.3390/jcm11164785)

Thank You very much for the recommendation but our study was conducted after the peak of covid and we started performing neuromuscular ultrasound in our center a little bit later. 

  1. Conclude the introduction with your study hypothesis.

The introduction was filled with the hypothesis: “We hypothesize that a specific pattern of nerve size and echogenicity might exists in the Lithuanian cohort. Also, nerve size probably has an association with anthropometric parameters”.

  1. Method: It better details the age range (line 64).

The age range was added: " Altogether, 125 healthy subjects with age range of 25-80 yrs were recruited for this study."

  1. Specifies whether patients enrolled in the study received a fee or it was all for free.

The text filled "Participants have not received any honorarium, participation was voluntary." 

  1. Specify the inclusion and exclusion criteria more precisely. considering patients without a peripheral nerve disorder, indicate exactly the neurological disorders enrolled, if possible, also express the data in a table.

The neurological disorders there: epilepsy, transient ischemic attack, stroke, and headache.

We presented disorders and the subjects’ number in brackets in results section: "Four subjects were treated for epilepsy, seven for transient ischemic attack, four because of stroke, and nine subjects investigated because of headache ". 

  1. Why did the second investigator evaluate only 20 subjects? I believe that all patients seen by the first should be reviewed by the second investigator as well!! The sentence is not clear, explain it.

Thank You very much for this insight. We based our numbers on study done previously  https://pubmed.ncbi.nlm.nih.gov/30593519/, doi: 10.1212/WNL.0000000000006856 . Mentioned study measured 30 subjects (total remeasured site number was 840). 920 different measurement sites were remeasured in our study by two investigators, and we kept this high number sufficient to implement reliable inter-ratio.

  1. Conclude the methods, before the statistical section, by stating exactly the endpoints of the study and therefore the hypotheses of the work.

Our hypothesis left the same as described in introduction part, so we have not repeated it avoiding not necessary information.

The endpoint was: “Therefore, the aim of our study was to establish reference values of CSA and echogenicity of peripheral nerves in a Lithuanian cohort of healthy adults in association with demographic and anthropometric data”.

9.Discussion: The objectives of the work are not clear. What advantage to the scientific literature might your study add?

The discussion additionally filled with: “Our study might have an important contribution for future studies analyzing if disparity exists according to different ethnic groups and different world regions. Also, our study might have important implications for further nerve size studies analyzing gender differences and body anthropometric parameters impact on nerve size and echogenicity ultrasound measurement results.”

  1. Conclude this section with limitations, strengths, future implications of your paper.

We concluded discussion with: “Our study has several limitations. Firstly, a part of our cohort were patients treated in the Neurology Department, so they were not strictly healthy subjects. However, the pathologies they were hospitalized for did not damage the peripheral nerves. Also, these subjects underwent neurological examination, which was normal. Secondly, we calculated nerve size correlations with various anthropometric data except for correlations with the length of limbs. Thirdly, both investigators had a relatively short period of experience in neuromuscular HRUS. However, the inter-ratio and intra-ratio reliability was excellent.

Strengths: We have analyzed a large cohort of healthy Lithuanian subjects and made a substantial number of measurements which allowed us to establish high-quality nerve size reference values.”

Thank You so much for all the comments.

Reviewer 2 Report

Comments and Suggestions for Authors

In the manuscript “Cross-Sectional Area and Echogenicity Reference Values for Sonography of Peripheral Nerves in the Lithuanian Population,” the authors try to provide reference data for nerve size and echogenicity for Lithuanian population. The manuscript is rigorously written, presents important data; however, several improvements should be done.

Abbreviations: Consider changing the abbreviation for the superficial fibular nerve to SFN as often seen in the literature.

Introduction:  The introduction is brief. I suggest that the authors expand it to explain the need for referential ultrasound values for peripheral nerve imaging.

Methods: US Examination: Please include the exact locations of the nerve measurements rather than: “were measured at specific sites according to NUP"

Echogenicity: The authors mention that quantitative image analysis was performed on native images converted into 8-bit images, then analysed automatically. Explain if the conditions for image scanning were the same for all patients. Additionally, specify if this protocol was previously used in similar studies. Also, please comment in the discussion whether there are already similar studies with histology serving as a basis for such assessments.

Results:  I appreciate that the authors focus on differences between males and females and the factors influencing the cross-sectional areas (CSAs) of peripheral nerves, including demographic data.  

If possible, consider condensing results into more tables, particularly in section “3.3. Nerve size correlation with demographic and anthropometric parameters,” where there is an abundance of text (and brackets).

The important thing missing within this evaluation is the fascicle differentiation (depiction). Do the authors maybe have any data regarding this?

Discussion: The authors state that evaluators had 1 or 2 years of experience in musculoskeletal US. It is imperative that authors find some article regarding the relevance of evaluator experience in performing musculoskeletal US on measuring the CSAs.

The second paragraph of the discussion does not add much value as it repeats the information from previous sections. Consider omitting this part or condensing it with the first paragraph. Furthermore, the info from previous studies is important but can be summarized and synthesized beyond merely repeating what has been found. Try to synthesize this info into more generalized principles that are in play when studying nerve CSA.

Additionally, please include more relevant and diverse literature to support the findings in discussion.

Comments on the Quality of English Language

Adequate.

Author Response

Dear reviewer,

Thank You so much for Your insights and notes. We found them very valuable. We are sending our corrections and addition information which was asked. 

  1. Abbreviations: Consider changing the abbreviation for the superficial fibular nerve to SFN as often seen in the literature. 

Superficial fibular nerve abbreviation was changed to SFN as recommended.

  1. Introduction:  The introduction is brief. I suggest that the authors expand it to explain the need for referential ultrasound values for peripheral nerve imaging.

Introduction was reorganized and filled with the additional information: 

" Reference measurements are needed in many areas of medicine, in order not only to make a more accurate diagnosis, improve treatment options, but also to collect data for scientific purposes, to analyze external factors that may have some influence. For the introduction of reference values, not only the specificities of different medical machines are important but also the experience of each researcher. "

  1. US Examination: Please include the exact locations of the nerve measurements rather than: “were measured at specific sites according to NUP".

 The note was corrected to: " Vagal nerve (VN) was measured at the level of the carotid triangle. Upper limb nerves: median (MN) and ulnar (UN) measurements were done at the middle of upper arm, elbow and middle of forearm (the MN was also measured at wrist and UN at Guyon’s canal), radial nerve (RN) was measured at spiral groove, superficial RN at the arcade of Frohse. Lower limb nerves: tibial nerve (TN) and fibular nerve (FN) were measured at popliteal fossa and TN was measured at ankle beneath the vascular arcade, sural nerve (SN) was measured next to the saphenous vein at the lateral ankle and in the calf. " 

  1. Echogenicity: The authors mention that quantitative image analysis was performed on native images converted into 8-bit images, then analysed automatically.

4a) Explain if the conditions for image scanning were the same for all patients.

4b) Additionally, specify if this protocol was previously used in similar studies.

4c) Also, please comment in the discussion whether there are already similar studies with histology serving as a basis for such assessments.

4a) Yes, all the conditions were used the same, as it was described in methodology " Ultrasound examination of peripheral nerves was performed using a Philips EPIQ 7 ultrasound machine (Philips EPIQ Diagnostic Ultrasound System, Bothell, WA, 2019) with a linear 4-18 MHz transducer (eL18-4, piezo elements 1920) in B mode. The main presets of parameters of ultrasonic scanning were as follows: initial scanning depth 2.5 cm, dynamic range was 77. Presets were constant during the entire study and only depth was adapted when needed. Focus was kept on the area of interest all the time. " Images were frozen and used for the echogenicity analyzis.

4b) Similar study was performed by Gamber et al, 2020 (https://pubmed.ncbi.nlm.nih.gov/32521091/, doi: 10.1111/jon.12717 ) unfortunately we could not compare results directly because there was a difference between used equipment.

4c) The discussion was filled with the citation of study analyzing sono-histology findings " As invasive diagnostic methods, such as nerve biopsy, become rarely used, sono-histological studies are of growing importance. It was shown that different sonographic patterns exist in acute and chronic phases of peripheral neuropathies."( https://pubmed.ncbi.nlm.nih.gov/35091803/, doi: 10.1007/s00415-022-10988-1 )

  1. If possible, consider condensing results into more tables, particularly in section “3.3. Nerve size correlation with demographic and anthropometric parameters,” where there is an abundance of text (and brackets).

​The biggest part of results in text in section 3.3. was removed, and results of nerve cross-sectional area correlations with anthropometric parameters supplied in Table Nr 4.

  1. The important thing missing within this evaluation is the fascicle differentiation (depiction). Do the authors maybe have any data regarding this?

Thank You so much for the idea. Unfortunately, we have not analyzed data from this perfective, it could be the aim for further our studies. 

  1. Discussion: The authors state that evaluators had 1 or 2 years of experience in musculoskeletal US. It is imperative that authors find some article regarding the relevance of evaluator experience in performing musculoskeletal US on measuring the CSAs.​

We have citated the article regarding the relevance of evaluator experience: https://pubmed.ncbi.nlm.nih.gov/28981150/ , doi: 10.1002/mus.25980.

8a) The second paragraph of the discussion does not add much value as it repeats the information from previous sections. Consider omitting this part or condensing it with the first paragraph. 

The part of second paragraph was mostly removed and other part condensed according to recommendations.

8b) Furthermore, the info from previous studies is important but can be summarized and synthesized beyond merely repeating what has been found. Try to synthesize this info into more generalized principles that are in play when studying nerve CSA. Additionally, please include more relevant and diverse literature to support the findings in discussion.​

The discussion was additionally filled with the other studies and meta-analysis finding. We tried to compare other study findings and summarize with ours:" When comparing our results to a a meta-analysis which analyzed 77 ultrasound studies on ulnar nerve CSA, we find that the measured ulnar nerve size is similar. However, we have found differences of ulnar nerve CSA according to sex that was not described by this meta-analysis. Our results match meta-analysis data showing that ulnar nerve size may depend on BMI, as we found that most ulnar nerve measurements positively correlated with BMI.

According to our study median nerve at the carpal tunnel (pisiformis level), forearm and at upper arm levels correlates with weight and BMI selectively in the female group. We have found that median nerve is smaller in women comparing to men along its entire length. This statement agrees with meta-analytical results which included 73 studies and stated that median nerve size values are higher in men then in woman [25]. We could not state that age influence nerve size in median nerve and this was a meaningful difference between meta-analysis and our results. We have noticed that median nerve is smaller along its length in the Lithuanian population comparing to the meta-analysis with the given size numbers, that leads that Lithuanian population wrist-to-forearm ratio of approximately is higher than ratio of 1.5 which was suggested in previous study [25].

A meta-analysis of CSA of tibial nerve at the ankle level of 3295 subjects showed a positive correlation between age and tibial nerve size which was contrary to our results. In our study we noticed that tibial nerve at the ankle level correlates with weight and BMI, and there is a much stronger correlations with both, weight and BMI, of the tibial nerve at the popliteal fossa.

Another meta-analysis analyzing 2695 lower limbs measurement [26] showed interesting results while comparing tibialis nerve size according to geographical regions. It was shown that healthy controls from Eastern Europe demonstrated larger tibial nerve CSA at ankle level while Europeans had smaller tibial nerve size [27]. The Lithuanian cohort has smaller tibial nerves at the ankle compared to the Eastern European cohort and corresponds to the results of the European cohort. Unexpectedly, tibial nerve CSA at popliteal fossa were larger in Lithuanian cohort comparing to both European geographical subgroups."

Thank You so much for all the comments.

Reviewer 3 Report

Comments and Suggestions for Authors

The present study came up with reference values of peripheral nerves in a specific population, both CSA and echogenicity. Overall, there is no concern regarding the methodology and the conclusions are entirely supported by the results. There are only a few comments to improve the paper. 

The discussion lacks comparison with previous studies, especially all those meta-analyses (Level I evidence). I would like to draw the authors’ attention to these studies in order to compare their findings with the ranges established from these meta-analyses.

The effect of age on nerve CSA: https://onlinelibrary.wiley.com/doi/full/10.1002/mus.27773

Ulnar nerve CSA: https://link.springer.com/article/10.1007/s40477-022-00661-8

Tibial nerve CSA: https://www.mdpi.com/2077-0383/12/19/6186, https://www.mdpi.com/1648-9144/58/12/1696

Median nerve CSA: https://systematicreviewsjournal.biomedcentral.com/articles/10.1186/s13643-018-0929-9

In addition to the above, there might be meta-analysis of other nerves too. 

The manuscript (font, table format, etc) should be formatted according to MDPI’s guidelines.

Comments on the Quality of English Language

No major concerns.

Author Response

Dear reviewer,

Thank You so much for Your insights and notes. We found them very valuable. We are sending our corrections and addition information which was asked. 

  1. The discussion lacks comparison with previous studies, especially all those meta-analyses (Level I evidence). I would like to draw the authors’ attention to these studies in order to compare their findings with the ranges established from these meta-analyses.​

The discussion was additionally filled with suggested articles: " When comparing our results to a a meta-analysis which analyzed 77 ultrasound studies on ulnar nerve CSA, we find that the measured ulnar nerve size is similar. However, we have found differences of ulnar nerve CSA according to sex that was not described by this meta-analysis. Our results match meta-analysis data showing that ulnar nerve size may depend on BMI, as we found that most ulnar nerve measurements positively correlated with BMI.

According to our study median nerve at the carpal tunnel (pisiformis level), forearm and at upper arm levels correlates with weight and BMI selectively in the female group. We have found that median nerve is smaller in women comparing to men along its entire length. This statement agrees with meta-analytical results which included 73 studies and stated that median nerve size values are higher in men then in woman [25]. We could not state that age influence nerve size in median nerve and this was a meaningful difference between meta-analysis and our results. We have noticed that median nerve is smaller along its length in the Lithuanian population comparing to the meta-analysis with the given size numbers, that leads that Lithuanian population wrist-to-forearm ratio of approximately is higher than ratio of 1.5 which was suggested in previous study [25].

A meta-analysis of CSA of tibial nerve at the ankle level of 3295 subjects showed a positive correlation between age and tibial nerve size which was contrary to our results. In our study we noticed that tibial nerve at the ankle level correlates with weight and BMI, and there is a much stronger correlations with both, weight and BMI, of the tibial nerve at the popliteal fossa.

Another meta-analysis analyzing 2695 lower limbs measurement [26] showed interesting results while comparing tibialis nerve size according to geographical regions. It was shown that healthy controls from Eastern Europe demonstrated larger tibial nerve CSA at ankle level while Europeans had smaller tibial nerve size [27]. The Lithuanian cohort has smaller tibial nerves at the ankle compared to the Eastern European cohort and corresponds to the results of the European cohort. Unexpectedly, tibial nerve CSA at popliteal fossa were larger in Lithuanian cohort comparing to both European geographical subgroups. "

  1. The manuscript (font, table format, etc) should be formatted according to MDPI’s guidelines.

We tried to follow and keep format which was suggested according to MDPI’s guidelines, changes in tables names additionally were done. 

Thank You so much for all the comments.

Round 2

Reviewer 2 Report

Comments and Suggestions for Authors

I appreciate all the effort that authors showed with the revised version of manuscript. The majority of the suggestions have been incorporated. I still believe the following parts should be enhanced or corrected before the acceptance:

-              The important limitation of this study should be added, as already previously mentioned:  this study was not assessing the nerve fascicles, but merely nerve CSAs and echogenicity.

-              Minor corrections for the Tables: please report the values using the same style: 0.001 or .001, but not both. Also: = and – signs used for explaining the abbreviations. Use just one not in mixed fashion. 

-              The discussion can still be slighly enhanced with more diverse publications covered. For instance, I suggest adding some transitional studies, establishing the connection between basic and clinical science in nerve CSAs. Perhaps consider preclinical studies as the following:  »CSAs of the tibial and fibular nerves near the popliteal fossa from this study were comparable to those observed in ex-vivo study assessing the fibular and tibial divisions of the sciatic nerve with HRUS and histological cross-sections. These findings implicates the relevance of such studies in establishing baseline values and bridging the gap between basic and clinical knowledge.« (10.1002/mus.28181)

-              The last paragraph should be: conclusions – I suggest to make it a little shorter, just key findings. Please also move the paragraph with limitations (and strengths) of this study before the conclusions.

Comments on the Quality of English Language

Adequate.

Author Response

Dear Reviewer,

Thank You one more time for all Your comments and efforts to help us to improve our manuscript. Here are the corrections:

1.The important limitation of this study should be added, as already previously mentioned:  this study was not assessing the nerve fascicles, but merely nerve CSAs and echogenicity.

The discussion part was filled with: „ And lastly, we have not assed the nerve fascicle’s size and the fascicle number what would have provided additional useful information next to CSA and echogenicity data”.

 2. Minor corrections for the Tables: please report the values using the same style: 0.001 or .001, but not both. Also: = and – signs used for explaining the abbreviations. Use just one not in mixed fashion

Notes were corrected in Table 4, and abbreviations under the tables.

  1. The discussion can still be slighly enhanced with more diverse publications covered. For instance, I suggest adding some transitional studies, establishing the connection between basic and clinical science in nerve CSAs. Perhaps consider preclinical studies as the following:  »CSAs of the tibial and fibular nerves near the popliteal fossa from this study were comparable to those observed in ex-vivo study assessing the fibular and tibial divisions of the sciatic nerve with HRUS and histological cross-sections. These findings implicates the relevance of such studies in establishing baseline values and bridging the gap between basic and clinical knowledge.« (10.1002/mus.28181)

The discussion was supplemented with suggested citation.

  1. The last paragraph should be: conclusions – I suggest to make it a little shorter, just key findings. Please also move the paragraph with limitations (and strengths) of this study before the conclusions.

Conclusions were moved below limitations and strengths and reorganized as follows: „In conclusion, reference values of nerve size and echogenicity of the Lithuanians presented for the first time. Our study showed that HRUS measurements of nerve size obtained from a large Lithuanian cohort of healthy subjects mostly are similar to those published previously but some differences have been found what could show that features specific to Baltic ethnicity might exist”.